# High-resolution spatiotemporal transcriptome mapping of tomato fruit development and ripening

Yoshihito Shinozaki[1], Philippe Nicolas[2], Noe Fernandez-Pozo[2], Qiyue Ma[2], Daniel J. Evanich[2], Yanna Shi[2,3], Yimin Xu[2], Yi Zheng [2], Stephen I. Snyder[1], Laetitia B.B. Martin[1], Eliel Ruiz-May[1], Theodore W. Thannhauser[4], Kunsong Chen[3], David S. Domozych[5], Carmen Catalá[1,2], Zhangjun Fei [2,4], Lukas A. Mueller[2], James J. Giovannoni[2,4] & Jocelyn K.C. Rose[1]

Tomato (*Solanum lycopersicum*) is an established model for studying fruit biology; however, most studies of tomato fruit growth and ripening are based on homogenized pericarp, and do not consider the internal tissues, or the expression signatures of individual cell and tissue types. We present a spatiotemporally resolved transcriptome analysis of tomato fruit ontogeny, using laser microdissection (LM) or hand dissection coupled with RNA-Seq analysis. Regulatory and structural gene networks, including families of transcription factors and hormone synthesis and signaling pathways, are defined across tissue and developmental spectra. The ripening program is revealed as comprising gradients of gene expression, initiating in internal tissues then radiating outward, and basipetally along a latitudinal axis. We also identify spatial variations in the patterns of epigenetic control superimposed on ripening gradients. Functional studies elucidate previously masked regulatory phenomena and relationships, including those associated with fruit quality traits, such as texture, color, aroma, and metabolite profiles.

[1] Plant Biology Section, School of Integrative Plant Science, Cornell University, Ithaca, NY 14853, USA. [2] Boyce Thompson Institute, Ithaca, NY 14853, USA. [3] Zhejiang Provincial Key Laboratory of Horticultural Plant Integrative Biology, Zhejiang University, Zijingang Campus, Hangzhou 310058, China. [4] U.S. Department of Agriculture-Agricultural Research Service, Robert W. Holley Center for Agriculture and Health, Ithaca, NY 14853, USA. [5] Department of Biology and Skidmore Microscopy Imaging Center, Skidmore College, Saratoga Springs, NY 12866, USA. Yoshihito Shinozaki, Philippe Nicolas and Noe Fernandez-Pozo contributed equally to this work. Correspondence and requests for materials should be addressed to J.K.C.R. (email: jr286@cornell.edu)

The fleshy fruits of angiosperms have evolved elaborate physiological and biochemical processes to promote animal attraction, consumption, and seed dispersal. These involve the interplay of numerous gene regulatory networks that are associated with diverse mechanisms of hormonal and epigenetic control throughout fruit ontogeny[1,2]. Tomato (*Solanum lycopersicum*) has emerged as the principal model to study fleshy fruit development and ripening, and to investigate the molecular bases of commercially important traits, including fruit set, size, texture, color, flavor, aroma, and nutritional quality[3,4], which support an industry with an estimated annual market value of > $55 billion[5]. Such targeted studies are supported by large-scale "omics"-based analyses of tomato fruit, including transcriptome, proteome, metabolome, and epigenome profiling initiatives[6–8].

Fruit anatomy is complex, with distinct interacting tissue and cell types, many of which undergo major changes in morphology during development, reflecting their functional diversification and specialization. However, most studies of tomato fruit biology to date have focused on only the pericarp, ignoring the various internal tissues that play diverse and potentially fundamental roles in fruit biology. As just one example, the expression patterns of families of genes that encode polysaccharide modifying proteins have been extensively studied in the pericarp, where they contribute to cell wall assembly during growth and disassembly during ripening[9]. However, the importance of such genes in processes such as the gelification of the locular tissue during ripening, or the structural degradation of other internal supporting tissues has not been addressed. Moreover, the common practice of homogenizing the isolated pericarp material prior to extracting analytes eliminates the possibility of acquiring cell-type-associated spatial information, and can dilute lower abundance molecules below the limits of detection[10,11]. Laser microdissection (LM) has been used to demonstrate the application of high-resolution transcriptome mapping of fruit tissue/cell-dependent genetic networks including in cuticle, cell wall, and carbohydrate metabolism[11,12], but such spatially resolved studies have only targeted fruit initiation or early fruit expansion. Other transcriptomic studies of plant organ/tissue development have resolved spatially or temporally specific patterns of gene expression at tissue or cell-level resolution, such as analyses of seeds[13] and roots[10]. However, current perspectives of the gene expression landscape in tomato fruit are of low spatial resolution and those related to a large portion of the fruit are missing entirely. Consequently, despite decades of research into the core molecular mechanisms underlying processes such as fruit expansion, softening, color change, and the accumulation of phytonutrients, the lack of holistic and high-resolution gene expression data remains a major barrier to advancing progress in understanding many areas of fruit biology.

We present here a comprehensive tomato fruit transcriptome atlas, incorporating global gene expression and co-expression data at a cell- or tissue-type level of resolution, including five pericarp cell/tissue types, whole pericarp and five fruit internal tissues, across the gamut of developmental stages, spanning early fruit growth and ripening. In addition, we take into account the fact that tomato fruit ripen along a latitudinal gradient, starting at the stylar, or blossom, end and spreading basipetally to the stem end[14], and examine transcriptional dynamics along this axis. We use these data sets to characterize the distribution and timing of gene regulatory and structural networks along spatial and developmental gradients, including functional evaluation of specific hormonal and epigenetic control points, thereby elucidating diverse aspects of fleshy fruit biology and quality traits that are typically indistinguishable in unseparated cells or tissues.

## Results

**Tissue or cell-type-based RNA-Seq of developing tomato fruit.** We generated transcriptome data by RNA sequencing (RNA-Seq) of LM derived, or hand-dissected, samples from six major fruit tissues (Fig. 1a and Supplementary Figs. 1a and 2a–d) and five pericarp cell/tissue types (Fig. 1b and Supplementary Fig. 2e–x), at the equatorial regions of fruit collectively spanning ten developmental stages (Fig. 1c). Although "seeds" are themselves complex reproductive organs including distinct progeny (embryos), they were categorized here as a fruit tissue, and at anthesis corresponded to unfertilized ovules. Fruit tissues were also isolated from different latitudinal sections at pre-ripe and early ripening stages: in the latter, ripening-related color change is observable in the pericarp at the stylar end (Fig. 1d and Supplementary Fig. 1b).

A summary of RNA-Seq data from a total of 483 samples is shown in Supplementary Data 1 and 2. The expression values, indicated by reads per million mapped reads (RPM) (Supplementary Data 3 and 4), showed high correlations (Spearman correlation coefficient (SCC) = 0.85–0.99) among biological replicates (Supplementary Data 5 and 6). Boxplots showed that the distribution of the normalized expression levels were comparable with respect to both cell/tissue type and developmental stages (Supplementary Fig. 3). A total of 24,506 and 20,732 genes were expressed (averaged RPM ≥1) in the six fruit tissues and five pericarp LM samples, respectively, in at least one of the tissue/cell types at some point during development. Similar numbers were expressed at individual stages, but in general, more were detected during fruit expansion (Supplementary Fig. 4). Almost half the genes were ubiquitously expressed in the six fruit tissues (10,328 genes; 42%) or five pericarp cell/tissue types (9842 genes; 47%) throughout development. Conversely, relatively few genes displayed tissue/cell-specific expression throughout development: the largest subsets were in seeds (1189; 4.9%) among the tissue samples and in the vascular tissue (950; 4.7%) among the pericarp cell/tissue types (Supplementary Fig. 5). This is consistent with the high degree of differentiation and functional specialization of tissues within the seed[13], and the multiple distinct cell types that form the vascular tissue.

Of the 20,732 genes expressed in the dissected pericarp cell/tissue-type samples, 19,494 were also detected in total pericarp samples, meaning that 1238 genes (6.4%) were not identified in the total pericarp samples. The expression levels of these 1238 genes detected only through LM were relatively low (Supplementary Fig. 6a). Among the 19,494 genes detected in the total pericarp, a total of only 5.0% (ranging from 0.2% in the collenchyma to 2.6% in the vascular tissue) showed cell- or tissue-type-specific expression (Supplementary Fig. 6b). In contrast, of the 1238 genes that were detected only through LM, 64% showed spatially specific expression (ranging from 2.5% in the collenchyma to 35% in the vascular tissue; Supplementary Fig. 6c). These data indicate that the use of high-resolution transcriptome sequencing allowed us to detect transcripts that were only present in low amounts in certain pericarp cell types, and that are overly diluted in total pericarp samples.

Principal component analyses (PCA) revealed a clear clustering of transcript profiles, corresponding to tissues or cell types in the equatorial regions of the fruit, based on the developmental stage (Fig. 1e, f). Expression profiles in seeds and vascular tissue were largely distinct from other tissues or cells at each stage, consistent with the relatively large number of corresponding specifically expressed genes (Supplementary Fig. 5). Locular tissue profiles showed a lower correlation with other tissues throughout ontogeny, indicating a more distinct developmental trajectory. The cell types of the pericarp, other than the vascular tissue, showed close correlation at 10 days post anthesis (DPA), while collenchyma and parenchyma maintained high correlation

throughout development. In contrast, profiles of the outer and inner epidermal layers were distinct after 20 DPA, suggesting that a program of developmental differentiation was initiated between 10 and 20 DPA. Gene expression patterns in both fruit tissues and pericarp cell types became more closely correlated after the pink (Pk) stage as ripening progressed. Notably, at the breaker (Br) stage, which is characterized by the initiation of multiple ripening phenomena, transcript profiles between the pericarp and placenta or locular tissue were particularly distinguishable compared to earlier stages. This indicated differences in the progression of ripening between external and internal tissues at this early ripening stage, as also evidenced by differences in pigmentation (Supplementary Fig. 1).

The transcriptome data were integrated into the Tomato Expression Atlas (TEA; http://tea.solgenomics.net/) database[15], which includes two- and three-dimensional anatomical visualization of tomato fruit internal structures obtained by light microscopy and computed tomography, respectively (Supplementary Fig. 7). The TEA enables the visualization of gene expression profiles across cell/tissue types and developmental stages in several formats, including pictographs, as exemplified in Fig. 2.

**Spatiotemporal insights into key fruit quality traits.** Decades of research have sought to identify genes underlying commercially important fruit quality traits, but have almost always been based on the analysis of bulk pericarp tissues. The many key molecular pathways that contribute to attributes, such as texture, flavor, aroma, and metabolite profiles, have therefore been characterized to some degree. However, as exemplified below, we found that our high-resolution spatiotemporal expression data, as well as a holistic consideration of all tissues, provided new perspectives of the timing and distribution of gene expression underlying those traits.

Fruit shelf life and textural changes during ripening have long been associated with modification of the cell wall, and particularly with the depolymerization of pectin polysaccharides[9]. Two genes involved in pectin degradation, a polygalacturonase (*PG2A*)[9] and

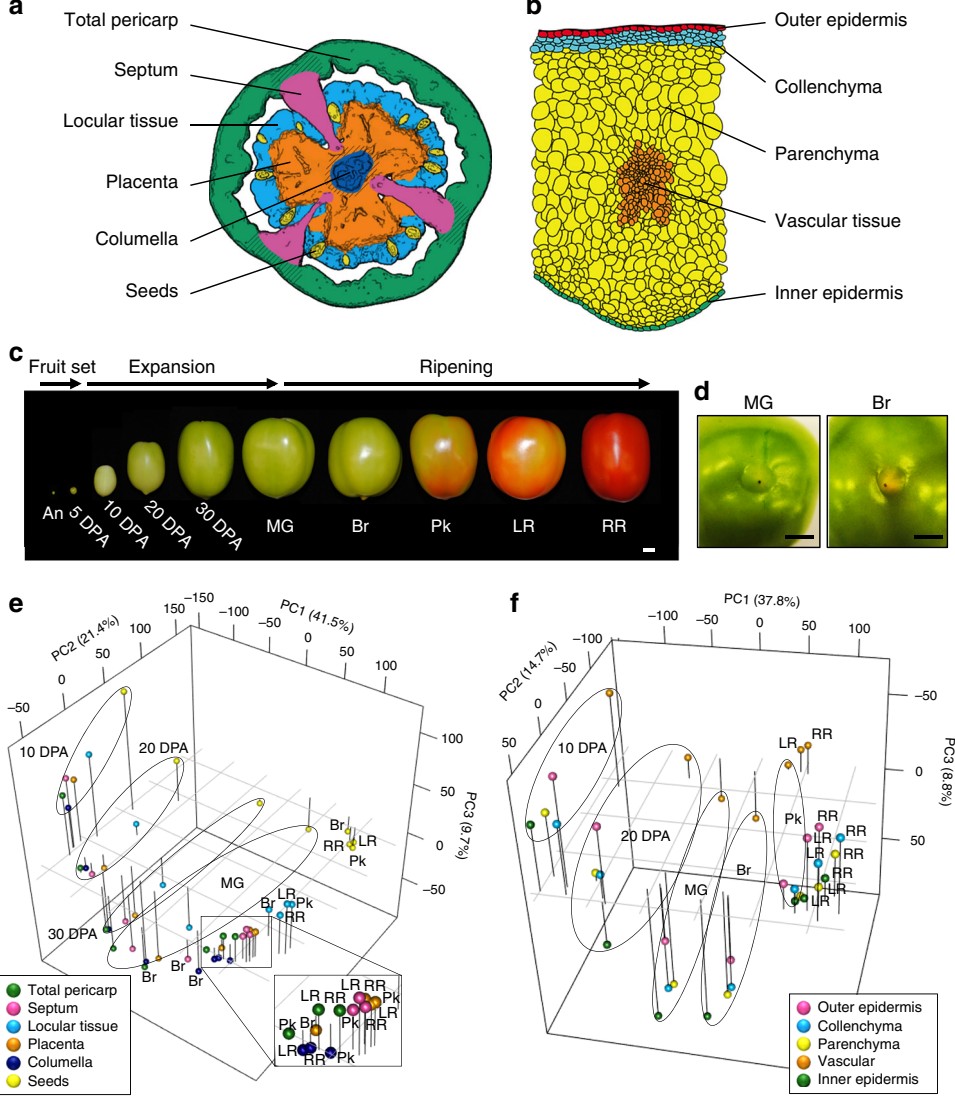

**Fig. 1** A tissue/cell-based transcript profiling of developing tomato fruit. **a** Traced image of six targeted fruit tissues. Shaded areas of the total pericarp and the placenta were not harvested. **b** Traced image of five pericarp cells. **c** Representative pictures of harvested fruit spanning ten developmental stages. **d** Representative pictures of the stylar end of MG and Br stage fruit. **e**, **f** Principal component analysis (PCA) of transcript profiles from six fruit tissues (**e**) and five pericarp cell/tissue types (**f**) through the fruit expansion and ripening. An, anthesis; DPA, days post anthesis; MG, mature green; Br, breaker; Pk, pink; LR, light red; RR, red ripe. Scale bars in **c**, **d** are 1 cm

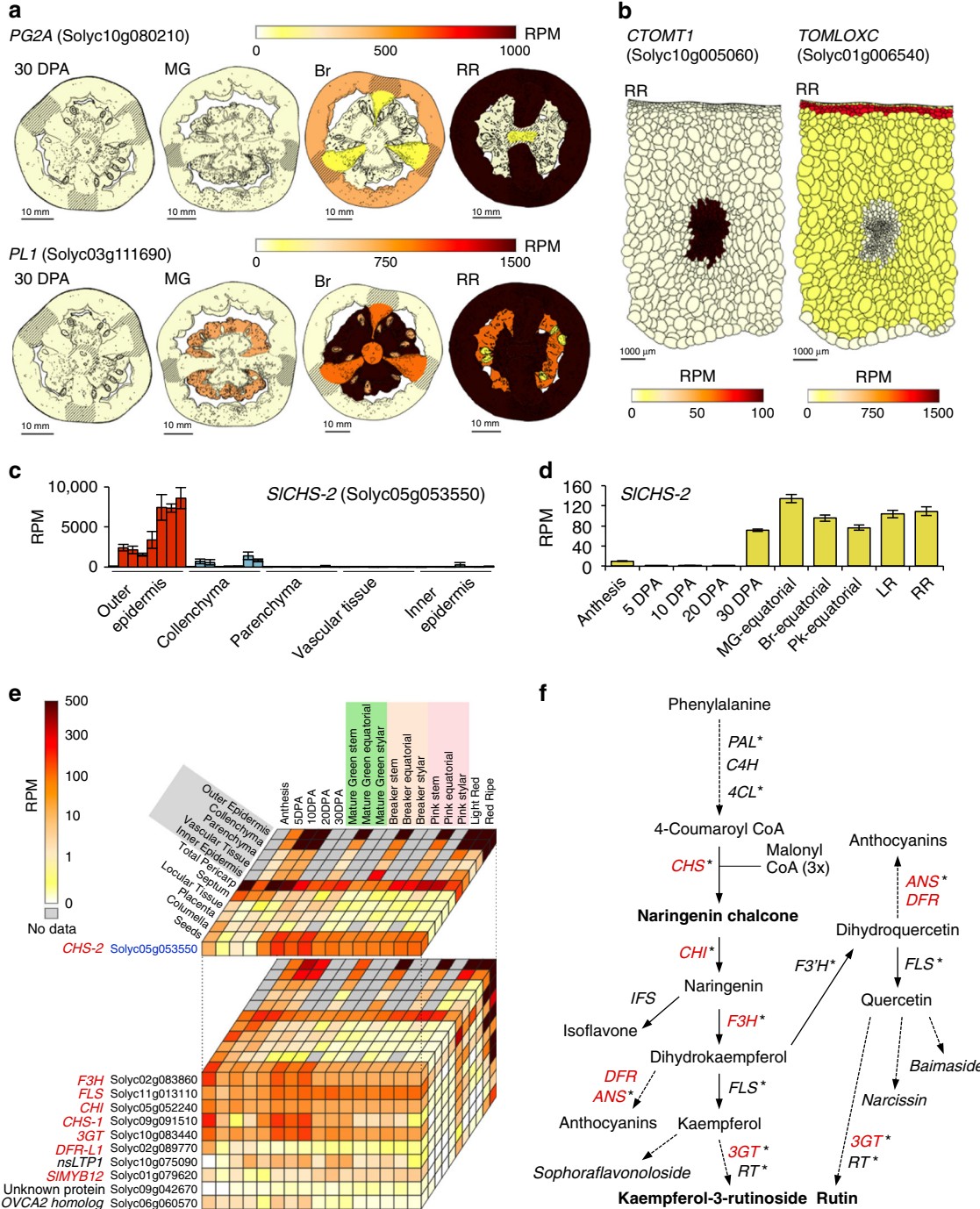

**Fig. 2** High-resolution transcript mapping of genes associated with fruit quality traits. **a**, **b** Tissue- or cell-based expression images of representative genes associated with fruit softening (**a**) and flavor (**b**). **c**, **d** The expression of *SlCHS-2* in the pericarp cell/tissue types (**c**) and seeds (**d**). Colored bars indicate mean RPM ± s.e.m. (*n* = 3 in **c** and 4 in **d**) in each cell/tissue type at each stage. Stages in **c** indicate 5 DPA, 10 DPA, 20 DPA, MG, Br, Pk, LR, and RR from left to right. **e** Expression cube showing the expression of *SlCHS-2* and co-expressed genes. The names of structural and regulatory genes in the flavonoid biosynthetic pathway are shown in red. **f** Overview of flavonoid biosynthetic pathway. Structural genes identified as co-expressed with *SlCHS-2* in **c** are shown in red. Genes transcriptionally activated by *SlMYB12*, based on previous studies[27,29], are indicated by asterisks. Major flavonoids, mostly flavonols, found in tomato fruit peel[27] and seeds[28] are shown in bold and italic, respectively. Pictograms in **a**, **b**, **e** were generated through the Tomato Expression Atlas (TEA) database. DPA, days post anthesis; MG, mature green; Br, breaker; PK, pink; LR, light red; RR, red ripe

a pectate lyase (*PL1*)[16], have been studied in detail, although not in internal fruit tissues, in order to understand the mechanistic basis of softening. Both are expressed during ripening, but are functionally distinct since suppressing the expression of *PL1*, but not of *PG2A*, substantially reduces the ripening-related loss of fruit firmness[16]. It may be that this reflects the sequential action

of the two pectinases, but we also uncovered distinct expression patterns, where *PG2A* was predominantly expressed in the pericarp and septum, whereas *PL1* showed earlier and abundant expression in the locule at the mature green (MG) stage, and then spread from internal to external tissues (Fig. 2a). Previous studies reported that most of the pectin in the locular tissue at MG or

earlier stages is de-esterified[17,18], and so is likely more susceptible to depolymerization by PL1. This contrasts with the pectins in other fruit tissues, including the pericarp and septum, where the ratio of de-esterified/esterified pectin increases later, during ripening[17,18]. Our results suggest not only that different modes of pectin breakdown and associated textural changes occur in different tissues, but that canonical ripening-related pectin disassembly initiates in internal tissues at a stage that is ostensibly "pre-ripe" based on external pericarp coloration.

Another well-studied aspect of ripening in tomato fruit is the change in volatile profiles that affect flavor and aroma[3,19]. Our high-resolution mapping revealed the pericarp cell/tissue-dependent expression of several genes involved in the biosynthesis of volatile compounds, which included the expression of *CTOMT1*, encoding a catechol-O-methyltransferase that produces guaia-col[20], exclusively in vascular tissues, and the predominant expression of *TOMLOXC*, encoding a 13-lipoxygenase essential for C5 flavor volatiles[21], in collenchyma cells (Fig. 2b and Supplementary Fig. 8). Similarly, we saw considerable spatial variation in the expression of metabolic genes involved in the accumulation of compounds that are nutritionally important or used as dietary supplements. For example, tomato fruit accumulate large amounts of gamma-aminobutyric acid (GABA), a non-proteinogenic amino acid, considered as a bioactive compound that has a range of benefits to human health[22,23]. We detected the highest expression of a glutamate decarboxylase (GAD) gene (*SlGAD3*), a key contributor to GABA levels[24], in the placenta during fruit expansion (Supplementary Fig. 9a), consistent with the accumulation pattern of GABA in tomato fruit[25]. Interestingly, *SlGAD3* showed predominant expression in the inner epidermis and vascular tissue in the pericarp (Supplementary Fig. 9b). The expression of *SlGABA-T1*, a GABA transaminase (GABA-T) involved in GABA catabolism in ripening tomato fruit, showed a broadly similar spatial expression profile to that of *SlGAD3*, although with a temporal delay (Supplementary Fig. 9).

**High-resolution gene co-expression networks**. Co-expression analysis using global transcript profiling allows the identification of functionally associated gene networks[26], and higher spatiotemporal data sets should result in more tightly defined network predictions. To test this idea, we first used a guide-gene approach and searched for genes that were co-expressed with a chalcone synthase gene, *SlCHS-2*. This encodes a rate-limiting enzyme involved in the biosynthesis of antioxidant flavonoid pigments, which accumulate in the tomato fruit peel during ripening[27]. *SlCHS-2* showed preferential expression in outer epidermal cells, which increased after the onset of ripening (Fig. 2c). Expression of *SlCHS-2* was also detected in seeds after 30 DPA (Fig. 2d), where various flavonol derivatives are known to accumulate[28]. The top 10 genes co-expressed with *SlCHS-2* (SCC = 0.74–0.85) (Fig. 2e) included six core genes encoding enzymes in the flavonoid biosynthetic pathway and a transcription factor (TF), *SlMYB12/COLORLESS FRUIT EPIDERMIS*, that regulates structural genes in the pathway[27,29] (Fig. 2f). Interestingly, the other three co-expressed genes (a non-specific lipid-transfer protein 1 (nsLTP1), which is also peel-enriched allergen[30], a protein of unknown function and an ovarian cancer-associated gene 2 protein (OVCA2) homolog) all have elevated expression in tomato fruit that ectopically express an *Arabidopsis thaliana* ortholog of *SlMYB12*[31,32], supporting the potential regulatory and functional associations among the identified genes.

As a complementary non-targeted approach to define the major gene regulatory networks throughout fruit development, we examined co-expressed gene sets reflecting spatiotemporal specificity, using weighted gene co-expression network analysis (WGCNA)[33]. A total of 43 modules were identified (M1–M43) based on the similar expression patterns, and associated with biological functions, including both known and new spatiotemporal aspects of tomato fruit ontogeny, using enriched gene ontology (GO) terms[34] and intramodular hub genes[33] (Supplementary Data 7–9, Supplementary Figs. 10–12 and Supplementary Note). Spatiotemporal data indeed provided gene regulatory network resolution and functional insights that could not be achieved via analysis of homogenized pericarp tissue alone.

**Predicting protein interactions in auxin signaling**. Global co-expression data also provide an opportunity to determine the conditions, tissues/cells, and the developmental stages in which functional protein–protein interactions may occur[35,36]. In this context, we investigated protein interactions associated with the plant hormone auxin, which plays an important role in fruit development[1]. Auxin signal transduction is mediated by protein–protein interactions between members of the auxin response factor (ARF) and Aux/IAA families[36–38]. Specific paired combinations of ARF and Aux/IAA proteins are necessary for the diversity of auxin responses in different tissues, developmental stages, and biological processes[39], and some tomato ARFs and AUX/IAAs have been functionally defined[40]. However, the specificity of most tomato ARF and Aux/IAA combinations has not been reported. We performed a pairwise expression analysis of 21 ARF and 24 Aux/IAA genes[37,38] across the different tissue/cell types and developmental stages. Among all the ARF-Aux/IAA combinations, the highest correlation (Pearson correlation coefficient (PCC) = 0.78) was found between *SlARF4* and *SlIAA15* (Fig. 3a), which showed relatively high expression during fruit expansion in all tissues other than seeds (Supplementary Fig. 13). SlARF4 is involved in the control of sugar metabolism and other ripening-related quality traits during fruit development[41], while SlIAA15 shows auxin responsive expression and plays multiple roles in vegetative development, including trichome formation, as well as in fruit set[42]. However, it is not known whether SlARF4 and SlIAA15 are interacting partners. A bimolecular fluorescence complementation (BiFC) assay with SlARF4 and SlIAA15 fused to the C- or N-terminal half of a yellow fluorescent protein (SlARF4-YFPC and SlIAA15-YFPN, respectively) resulted in a strong positive fluorescence in the nucleus (Fig. 3b), confirming physical interaction in planta. Co-introduction of YFPC fused to SlARF3, which lacks essential binding domains to Aux/IAAs[43,44] and belongs to a phylogenetic sister group of SlARF4, with SlIAA15-YFPN was used as a negative control, and resulted in no fluorescence. A significant barrier to elucidating gene regulatory networks is the large size of transcriptional regulator families, and the consequent multiplicity of potential interactions between proteins from these families. Our ability to predict and then confirm the interaction between SlARF4 and SlIAA15 proteins, both members of large families, illustrates how the high-resolution spatiotemporal expression information can facilitate the prediction and experimental validation of interacting protein partners.

**Cell-type-based variation in carbon metabolism-related genes**. One of the most studied aspects of tomato fruit biology is the acquisition and metabolism of carbon compounds and the factors that influence its function as a sink organ[45]. Synthesis, metabolism, and accumulation of sugars and organic acids are key determinants of nutritional and commercial value. While many of the genes, proteins, and enzyme activities associated with photosynthetic carbon fixation and storage have been characterized in terms of their developmental patterns in tomato fruit, little is known at the cell/tissue level of resolution.

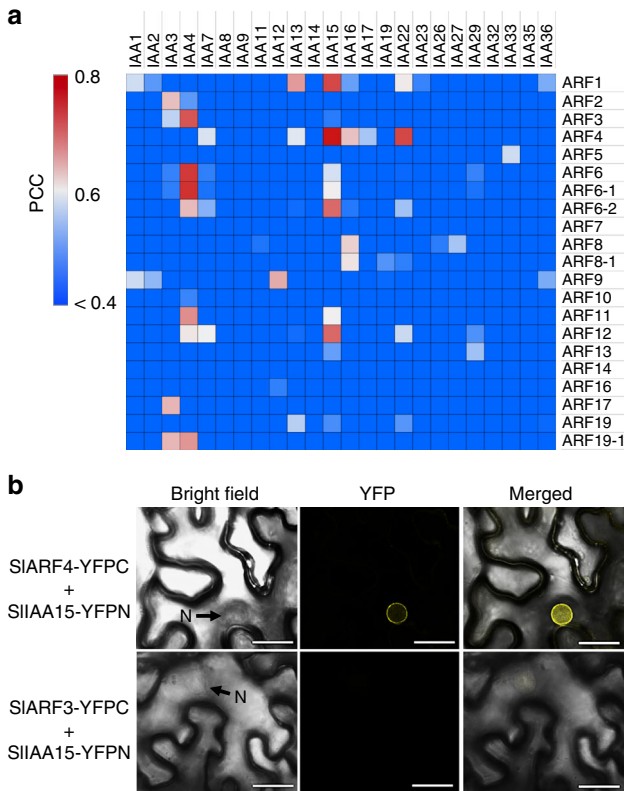

**Fig. 3** Identifying protein interactions between spatiotemporally co-expressed ARF and Aux/IAA auxin signaling associated proteins. **a** Heat map showing pairwise correlation scores between auxin signaling related *ARF* and *Aux/IAA* genes across the analyzed fruit tissues and cells during development. Pearson correlation coefficient (PCC) values were calculated for each pair of *ARF-Aux/IAA* genes. **b** SlARF4 interacts with SlIAA15 in planta. BiFC analysis in *N. benthamiana* leaf epidermal cells expressing full length of *SlARF4* and *SlIAA15* fused to the amino (YFPN) or carboxyl (YFPC) portions of YFP. Yellow fluorescence indicates interaction between SlARF4-YFPC and SlIAA15-YFPN. No clear interaction was detected between SlARF3-YFPC and SlIAA15. N, Nucleus. Scale bars: 25 μm

During early tomato fruit development, fixed carbon is imported as sucrose from source leaves and then converted to starch for storage[45,46]. Among the pericarp cell types, we found that several genes encoding key enzymes involved in starch biosynthesis, including ADP-glucose pyrophosphorylase (AGPase) and starch-branching enzyme, showed particularly abundant expression in the parenchyma (Fig. 4a). Accordingly, transmission electron microscopy (TEM) analysis revealed that at 5 DPA, the parenchyma cells adjacent to the inner epidermis predominantly contained plastids harboring grana with stacked thylakoids and large starch granules, indicating an intermediate structure between chloroplasts and starch-storing amyloplasts (Fig. 4b, c). The starch granules in the plastids of the parenchyma developed further and the grana structures were disrupted at 10 DPA (Fig. 4d). This observation was consistent with previous reports that starch accumulates predominantly in the inner layers of the pericarp[12,46], and here we found that this occurs specifically in the parenchyma cells, and not in the inner epidermis.

Although young green tomato fruit are photosynthetically active, several studies have suggested that fruit photosynthesis is not essential for primary carbon metabolism and growth, but may be important for seed development[45,47,48]. Gene co-expression modules associated with the pericarp inner epidermis (M33), or

both the inner epidermis and parenchyma (M32), showed significant enrichment of several photosynthesis-related GO terms and hub genes (Supplementary Data 8 and 9 and Supplementary Fig. 12). Consistent with this, we found that among the pericarp cell/tissue types, the inner epidermis showed the most substantial expression of genes related to photosynthetic light reactions, as well as several key genes of the Calvin cycle, including those encoding small subunits of ribulose 1,5-bisphosphate carboxylase/oxygenase (RuBisCO) (Fig. 4e) and glyceraldehyde-3-phosphate dehydrogenase (GAPDH), during fruit expansion (Supplementary Fig. 14). This pattern was also evident in parenchyma cells, but to a lesser extent.

The predominance of photosynthesis-related genes in the inner epidermal cell layer, which is shielded from light by the overlying tissues, is somewhat counterintuitive. In the absence of light, proplastids do not differentiate to chloroplasts, but rather develop into etioplasts that lack chlorophyll[49]. However, TEM analysis confirmed that the inner epidermis at 5 DPA contains chloroplasts with grana, rather than typical etioplasts (Fig. 4b, f), and that the grana structures were further developed at 10 DPA (Fig. 4g). We propose that despite its location, the pericarp inner epidermis of tomato fruit is most likely photosynthetically active, and may play a role in the photoassimilation of $CO_2$ from the fruit locular cavity, as has been suggested to be the case in the dry, dehiscent pea fruit/pod[50]. The high expression of photosynthetic genes in the inner epidermis may also contribute to maintaining sufficient $O_2$ in the locular cavity for successful seed development, since such a role for $O_2$ has been suggested in pepper (*Capsicum annuum*) fruit[51]. The cell-type-dependent gene expression profiling therefore highlights clear spatial variation in central carbon metabolism across the pericarp and provides a platform for investigating mechanisms by which sugars traffic between, and are distributed among, the constituent cell types.

**Spatiotemporal regulation of tomato fruit ripening.** The onset and progression of ripening in tomato is typically associated with changes in the external color of the pericarp, reflecting carotenoid and flavonoid pigment accumulation. Indeed, the Br ripening stage is so-called due to the "break" in external color (Fig. 1c, d). However, prior to this event, changes reminiscent of ripening occur in the fruit inner tissues (Supplementary Fig. 1), including pigment accumulation in some genotypes, as well as gelification of the locular tissue associated with substantial pectin degradation[52] and induction of *PL1* (Fig. 2a). We found that the WGCNA co-expression module M6 contained a number of genes with well-established functions in various aspects of ripening (Supplementary Data 7). These included the TFs *RIPENING INHIBITOR* (*RIN*; Fig. 5a) and *NON-RIPENING* (*NOR*), which act as key ripening regulators[8,53], *PL1* and an expansin (*LeEXP1*), which are associated with softening[9,16], two carotenoid biosynthetic genes, encoding phytoene synthase (PSY1) and carotenoid isomerase (CRTISO)[54], and the ripening-associated gene E8[55]. Expression of these genes, as well as most of those in M6, starts in the locular tissue and then radiates out to the placenta and other tissues, but does not occur in seeds (Supplementary Fig. 10). This indicates that the ripening program initiates in internal tissues before the Br stage, although the earliest molecular changes that initiate and define, de facto ripening have not yet been identified.

We found that *RIN* forms a central hub gene in M6 (Supplementary Data 9), suggesting that it acts as a major regulator of co-expressed genes in this module. Gene sets that were reported to be positively regulated by, and/or direct targets of, RIN[56,57] were particularly enriched in module M6 (Supplementary Fig. 15a) and to a lesser degree in M18 and M39. To identify new ripening regulators based on these module

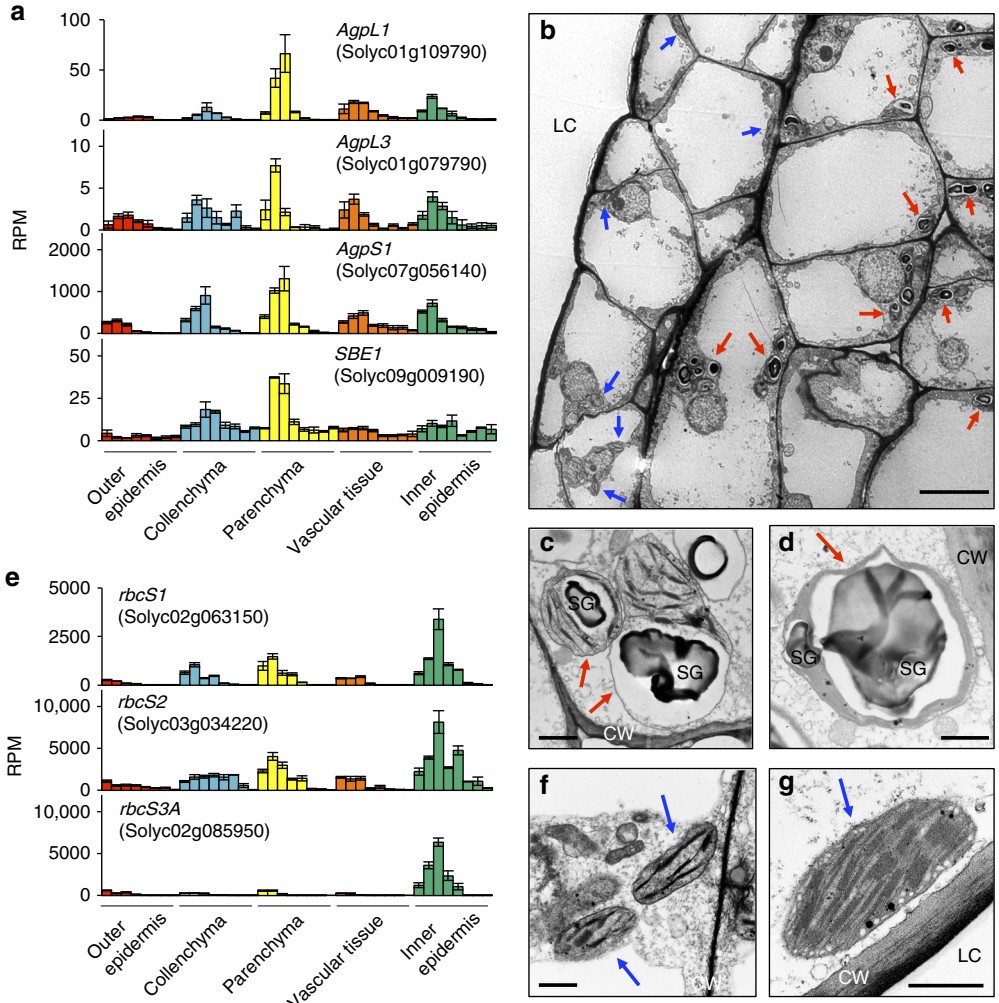

**Fig. 4** Cell-type-dependent differentiation of plastids in the pericarp. **a**, **e** Expression of genes associated with starch biosynthesis (**a**) and photosynthesis (**e**). Colored bars indicate mean RPM ± s.e.m. (n = 3) in each cell type at each stage (left to right; 5 DPA, 10 DPA, 20 DPA, MG, Br, Pk, LR, and RR). **b**–**d**, **f**, **g** Transmission electron microscopy (TEM) images show plastids in pericarp inner cell layers (**b**), parenchyma adjacent to inner epidermis (**c**, **d**) and inner epidermis (**f**, **g**). Representative images from 5 DPA (**b**, **c**, **f**) and 10 DPA (**d**, **g**) fruits are shown. Plastids containing starch granules are indicated by red arrows. Chloroplasts without large starch granules are indicated by blue arrows. The grana with stacked thylakoids in plastids form linear structures with a higher electron density. DPA, days post anthesis; CW, cell wall; LC, locular cavity; SG, starch granule. Scale bars: 10 μm in **b** and 1 μm in **c**, **d**, **f**, **g**

assignments, we focused on another hub gene (Supplementary Data 9) encoding a GRAS family TF, *SlGRAS38* (Solyc07g052960)[58], from M6, which has been reported to be a direct target of RIN[56,57] (Fig. 5a). We generated six independent *SlGRAS38*-RNAi silenced tomato lines in cultivar Ailsa Craig that had reduced *SlGRAS38* expression (Fig. 5b and Supplementary Fig. 16a), of which two (*gras-13* and *gras-20*) were chosen for more detailed characterization. All transgenic and WT Ailsa Craig untransformed control lines were grown together and fruit were tagged at the Br stage. To identify transcriptomic changes resulting from *SlGRAS38* repression, we performed RNA-seq analysis of *gras-13*, *gras-20*, and WT pericarp samples at the Br stage. A large proportion of the genes with reduced expression in both transgenic lines, compared to WT with ratio ≤0.5 (FDR-adjusted *p* value <0.05, exact test using the edgeR software package), showed significant overlap with M6 and adjacent M39 assigned genes (Supplementary Data 10 and Supplementary Figs. 12 and 15b), indicating a regulatory function for *SlGRAS38* in these ripening-related gene networks. The *SlGRAS38*-silenced plants produced fruit with significantly lower content of the predominant carotenoid, lycopene, in the pericarp of both red

ripe (RR) and over-ripe stage fruit (7 and 15 days after Br stage, respectively; Fig. 5c and Supplementary Fig. 16b), although the impaired coloration was not as severe as in the *rin* mutant (Fig. 5d). Production of the ripening-related hormone ethylene was also lower in the *SlGRAS38*-silenced lines (Fig. 5e). Of the 32 genes that were downregulated in both *gras-13* and *gras-20*, 10 are repressed in the *rin* mutant[56] and 3 are direct RIN target genes[57], but this subset includes no known ripening regulators or structural genes involved in carotenoid or ethylene metabolism (Supplementary Data 10). These results indicate that *SlGRAS38* represents a newly discovered component of the regulatory cascade downstream of *RIN*, and may coordinate ripening processes associated with carotenoid and ethylene metabolism that have not yet been elucidated.

**Latitudinal gradients of gene expression in fruit tissues**. In parallel with the gradients of ripening-related gene expression from the interior to the exterior of the tomato fruit, a notable feature of ripening in tomato is its spatial progression along the latitudinal axis. This is not to be confused with the gradient in

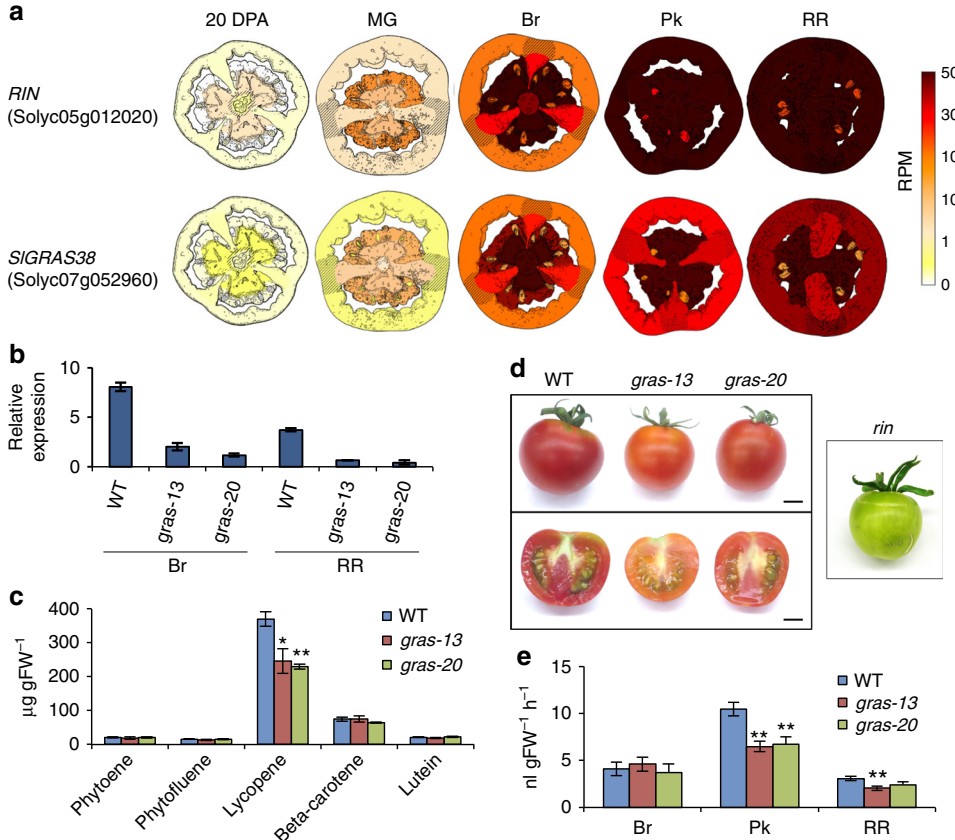

**Fig. 5** The role of *SlGRAS38* in fruit ripening regulation. **a** Tissue-based expression images of two M6 hub TF genes, *RIN* and *SlGRAS38* (Solyc07g052960). **b** Expression of *SlGRAS38* in Br and RR transgenic fruit from two independent *SlGRAS38*-RNAi lines and WT normalized to *RPL2*. **c** Carotenoid levels of RR fruit at 7 days after Br from *SlGRAS38*-RNAi lines and WT. **d** Representative fruit of WT, *SlGRAS38*-RNAi, and *rin* mutant plants. Fruit of WT and *SlGRAS38*-RNAi lines at 4 days after Br and that of *rin* at a stage corresponding to RR in WT are shown. Scale bars are 1 cm. **e** Ethylene production in *SlGRAS38*-RNAi and control fruit. Values are mean ± s.e.m. (*n* = 3 in **b**, **c**, and 7 in **e**). Asterisks in **c**, **e** indicate significant differences between WT and each *SlGRAS38*-RNAi line (**$P < 0.01$, *$P < 0.05$; Student's *t* test)

plastid accumulation regulated by the *UNIFORM* gene, which operates separately from the ripening gradient[14]. Previous analyses, focused only on the ripening pericarp, revealed that several genes associated with ethylene biosynthesis and responses show a latitudinal gradient of expression in RR, but not in Br stage fruit, although core ripening regulators, including *RIN* and *NOR*, do not show ripening-related expression gradients[14]. Here, we identified differentially expressed genes (DEGs) with ratio ≥1.5 or ≤0.66 (FDR-adjusted *p* value <0.05, exact test using edgeR) in various fruit tissues comparing stem-region and stylar-region (total pericarp, septum, locular tissue, placenta, and seeds), or stem-region versus equatorial-region (columella) during early ripening (Supplementary Data 11 and Supplementary Fig. 17a). Most DEGs showed differential expression in only one or two tissues at each stage (Supplementary Fig. 17b), suggesting the presence of highly tissue-dependent transcript profiles associated with the ripening gradient. The largest numbers of tissue-specific DEGs were found in the columella at the Br and Pk stages (Supplementary Fig. 17a), including several ethylene-biosynthetic or -responsive genes (*ACS2*, *ACS4*, and *E8*) and *RIN*, which showed higher expression in the equatorial-region than in the stem-region (Supplementary Fig. 17c). *NOR*, which acts upstream of *RIN*[8], showed a gradient of expression in the MG placenta and columella at the transition from maturation to ripening, consistent with a role in establishing the ripening gradient in internal tissues. Among the DEGs shared in three or more tissues, we found several genes associated with another ripening-related

hormone, abscisic acid (ABA), including key ABA metabolic genes (*NCED1* and *CYP707A2*)[59] and an ABA receptor gene (*PYL1*)[60] (Supplementary Fig. 17d). Notably, several genes associated with auxin, which has been proposed to inhibit tomato fruit ripening[61], showed consistently higher expression in the stem-region in all tissues (e.g., auxin-responsive *SAUR37* and *IAA3*), or specifically in the placenta and columella at particular stages (e.g., the auxin transporters *LAX1* and *LAX2*) (Supplementary Fig. 17e). Collectively, this analysis highlighted the spatial variation along the fruit latitudinal axis of interacting hormone networks and TF-mediated signaling.

**Spatial variation in epigenetic regulation of ripening.** In addition to the network of TFs and hormonal regulation, epigenetic modifications, involving changes in DNA methylation, have also been shown to regulate tomato fruit ripening[2,7,8]. We noted that the ripening-associated module M6 included *SlDML2* (Solyc10g083630) (Supplementary Data 7), a DEMETER-like DNA demethylase gene that has been shown to control the loss of 5-methylcytosine in promoters of key ripening regulators, such as *RIN*, in the fruit pericarp during ripening, or at earlier stages[62,63]. This suggested possible tissue-dependent regulation of demethylation at promoters of ripening regulators in M6 that might modulate their spatiotemporal expression pattern during fruit ripening. To test this hypothesis, we targeted *RIN*, a central hub gene in M6. As the *RIN* promoter is a known target of SlDML2-

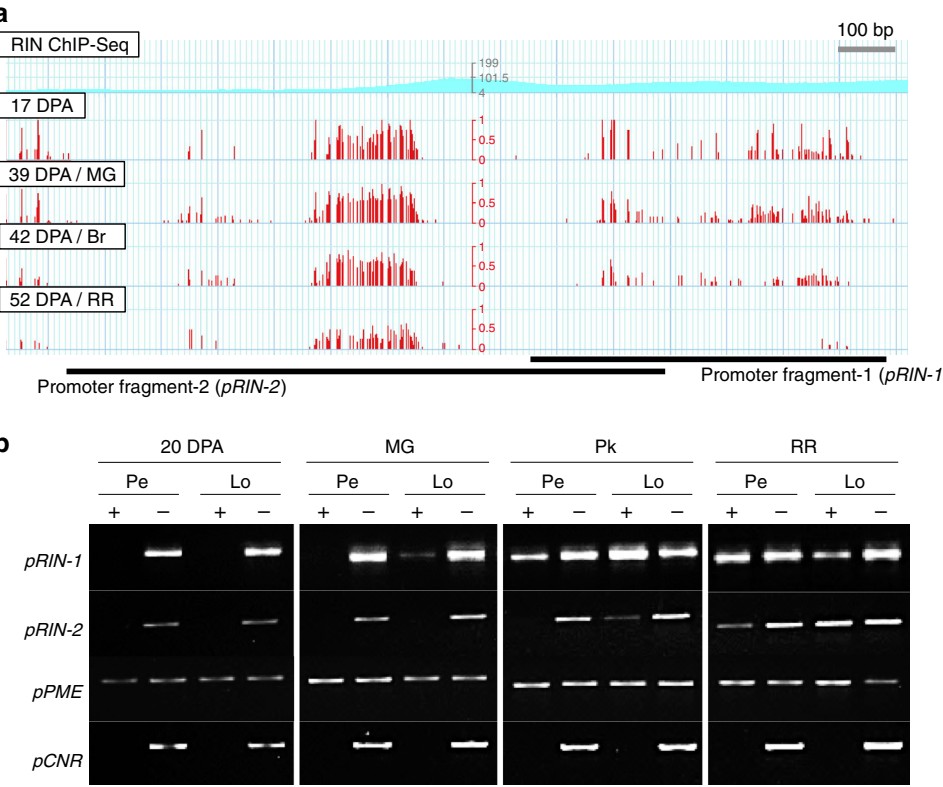

**Fig. 6** Differential ripening-related demethylation of the promoter of *RIN* between pericarp and locular tissue. **a** RIN self-binding sites and differentially methylated regions in the *RIN* promoter, based on analyses of pericarp samples (available at ted.bti.cornell.edu/epigenome). ChIP-seq plot depicts position of RIN binding, with peak height representing the number of reads covering each site (i.e., read depth). Red bars indicate cytosine methylation ratios at the equivalent position. Two promoter fragments analyzed by McrBC-PCR are indicated by black bars. **b** McrBC-PCR analysis of two *RIN* promoter fragments in pericarp (Pe) and locular tissue (Lo) at four stages spanning fruit expansion and ripening. A total of 0.5 µg genomic DNA was digested by McrBC (NEB) with GTP (+), or without GTP (−) as a negative control. An unmethylated region of the *PME* (Solyc03g123630) promoter is included as a positive control, and a highly methylated region of the *CNR* (Solyc02g077920) promoter showing complete digestion of methylated DNA. DPA, days post-anthesis; MG, mature green; Br, breaker; Pk, pink; RR, red ripe)

mediated demethylation[62,63], it was used here as an indicator of more general locule demethylation. An McrBC-PCR analysis was conducted to examine the DNA methylation status of two regions of the *RIN* promoter (*pRIN-1* and *pRIN-2*; Fig. 6a) spanning RIN self-binding sites. These regions were also shown to be differentially methylated in pericarp samples during ripening[2]. Ripening-associated demethylation of both regions was observed in locular tissue prior to demethylation in the pericarp, at the MG (*pRIN-1*) or Pk (*pRIN-2*) stages (Fig. 6b). These results illustrate a pattern of tissue-specific epigenetic changes, and suggest that differences in promoter methylation levels across tissues contribute to spatiotemporal differences in the expression of key ripening regulators and phenomena, with the ripening program initiating in the locule prior to the pericarp.

## Discussion

We present a global analysis of the tomato fruit transcriptome through tissues, cell types, development, and fruit topography. A total of 24,660 unique genes were detected in at least one cell/tissue and stage, corresponding to 71% of predicted protein-coding genes in the tomato genome. The high sensitivity of the LM-based RNA-Seq approach enabled the detection of low-abundance genes and their spatial assignment to the specific cell types of the pericarp. Moreover, the detection of 1238 genes in LM-derived samples, but not in those from total pericarp (Supplementary Fig. 6) underlines the considerable potential value of cell-type-specific profiling for inspiring functional studies. We found that the fruit internal tissues, which are rarely considered in studies of tomato fruit biology, collectively express 2464 genes that were not detected in the pericarp. Taken together with the cell/tissue-type-specific profiling, these gene expression data sets substantially enhance both the qualitative and quantitative resolution of the fruit transcriptome landscape, and provide a large reservoir of information for the development of new hypotheses.

In this study, we also describe several examples of data sets that elucidate spatiotemporal regulatory networks associated with various stages of fruit ontogeny, including mechanisms that control cell wall metabolism, as well as the production of volatile aroma compounds, GABA and carbohydrates. In addition, we show how these data can be used to predict protein interactions, and present an example of how this can be confirmed experimentally. Collectively, the data provide a platform for a broad range of studies to resolve the specialized functions of tissue/cell types and biological processes in fruit. It should be noted that understanding correlations, or lack thereof, between levels of transcripts and their corresponding proteins is fundamentally important for any systematic interpretation of complex molecular networks[64]. The high spatial resolution gene expression data set may enable more precise prediction of quantitative transcript-protein associations, although correlations can vary depending on the protein function, developmental stage, and cell/tissue type, as shown in metazoans[65,66] and plants[67,68].

High-resolution expression analyses revealed complex programs that were coordinately regulated across cell/tissue types and the developmental stages. These included latitudinal

expression gradients of hormonal genes associated with ethylene, ABA and auxin through the external and internal tissues, which likely play key roles in governing the ripening gradient (Supplementary Fig. 17). Another characteristic ripening-related expression program showing dynamic change across multiple cells/tissues was exhibited by co-expression module M6, members of which showed gradients of expression, initiating in internal tissues then radiating outward as maturation progressed (Supplementary Fig. 10). A previously uncharacterized TF from the M6 module, *SlGRAS38*, was shown to have a function in ripening regulation (Fig. 5). The expression correlation of *PL1* and *LeEXP1* in this module suggests that they contribute to locular gelification prior to the cell wall disassembly in the pericarp that is a hallmark of fruit softening. Additionally, several key ripening regulators, represented by a central hub, *RIN*, were found in this module, suggesting a spatial origin of tomato fruit ripening in the locular tissue. This was further supported by earlier ripening-associated change in the epigenetic state of the *RIN* promoter in the same tissue (Fig. 6). An interesting possibility is that the seeds, which are embedded within the locule, provide a ripening signal that is transmitted through RIN. It is intriguing that parthenocarpic (seedless) fruit exhibit ripening-related changes, but they are delayed[69], consistent with a role for the seed in influencing ripening phenomena.

In summary, we present a high-resolution atlas of the tomato fruit transcriptome that provides new molecular insights into development and ripening. We have illustrated the value of these data sets to investigate diverse aspects of fleshy fruit biology, with examples selected to elucidate commercially important traits, such as texture, flavor and aroma, color, and metabolite composition, as well questions related to the orchestration of transcriptional, hormonal, and epigenetic regulatory mechanisms. These data, which are accessible via the public TEA database, will facilitate future hypothesis development and research in tomato, and represent a model for more precise molecular characterization of additional species and developmental programs.

## Methods

**Plant materials and tissue preparation**. Tomato (*S. lycopersicum* cv M82) plants were grown in a greenhouse at Cornell University (Ithaca, NY, USA) with a light/dark photoperiod of 16/8 h. Soil moisture sensors (MAS-1, Decagon Devices, Pullman, USA) were used to monitor pot volumetric water content (VWC) and to regulate an irrigation system to maintain 40–50% VWC. Flowers at anthesis (0 DPA) were tagged and manually pollinated and ovaries at anthesis were harvested immediately prior to the point when they would otherwise be self-fertilized. Expanding fruit were harvested at 5, 10, 20, or 30 DPA. Mature and ripening fruit were harvested based on the tomato color chart "USDA Visual Aid TM-L-1" (USDA Agricultural Marketing Service, 1975) as follows: MG stage (full-size green fruit, approximately 39 DPA), Br stage (approximately 42 DPA, definite break in color from green to tannish-yellow with less than 10% of the surface), Pk stage (50% pink or red color, approximately Br + 2 days), light red (LR) stage (100% light red, approximately Br + 4 days), red ripe (RR) stage (approximately breaker + 8 days with full red). Ovary and fruit samples were harvested from 60 randomly selected individual plants at the same time each day.

The six major fruit tissues, consisting of pericarp, septum, locular tissue, placenta, columella, as well as seeds (categorized here as a tissue-type), were collected by hand dissection from fruit equatorial sections, snap-frozen in liquid nitrogen, and kept at −80 °C until further analysis. For the MG, Br, and Pk stages, fruit tissues (with the exception of the columella) were also harvested from stem- and stylar-region latitudinal sections (see Supplementary Fig. 1b). At least 12 fruit were harvested for each biological replicate. Due to their small size, laser microdissection (LM) was used to harvest these six tissues at anthesis, as well as locular tissue, placenta, and seeds at 5 DPA. The five pericarp cell/tissue types (outer epidermis, collenchyma, parenchyma, vascular tissue, and inner epidermis) were harvested by LM from equatorial sections of fruit at 5 DPA, 10 DPA, 20 DPA, MG, Br, Pk, LR, and RR stages. Samples for LM were fixed, cryoprotected, embedded in optimal cutting temperature (OCT) medium, and frozen in molds using liquid nitrogen, whereas those collected at Pk, LR, and RR were directly embedded in OCT without fixation/cryoprotection, to preserve their integrity[70].

**Laser microdissection**. Sections of embedded tissues for the LM samples were mounted on adhesive-coated glass slides using a CryoJane transfer system (Instrumedics)[70], except the anthesis and 5 DPA samples, which were mounted on PET-membrane 1.4-μm slides (Leica). Section thickness was set at 12 μm for membrane slides and 16 μm for glass slides. Approximately 25 μm surface layers of the placenta at anthesis and 5 DPA were collected as prospective locular tissue (Supplementary Fig. 2c, d). Specimen areas ranging from $6.8 \times 10^5$ to $109.1 \times 10^6$ μm$^2$ were collected using a Zeiss PALM MicroBeam laser microdissection system, pooled from sections of at least three independent fruit/pericarp samples and stored at −80 °C.

**RNA isolation and sequencing**. Total RNA from hand dissection samples was isolated using an RNeasy Mini Kit (Qiagen). For LM samples, total RNA was isolated using an RNeasy Micro kit (Qiagen), and in vitro transcription was performed using a TargetAmp two-round aRNA amplification kit (Epicentre). Strand-specific libraries were constructed[71] and sequenced on a HiSeq2500 system (Illumina) from a single end for 100 cycles.

**Read mapping and transcript profiling**. Raw RNA-Seq reads were processed using Trimmomatic[72] to remove adapter and low-quality sequences, and then aligned to the ribosomal RNA database[73] using Bowtie[74] and the mappable reads discarded. The resulting high-quality cleaned reads were aligned to the tomato Heinz genome[75] reference SL 2.50 with gene models iTAG2.4 (https://solgenomics.net/organism/Solanum_lycopersicum/genome) using HISAT 2[76]. Following alignments, raw counts for each tomato gene were normalized to reads per million mapped reads (RPM). Pairwise Spearman correlation coefficient (SCC) values between biological replicates were calculated with log$_2$-transformed RPM values (i.e., log$_2$ [RPM + 1]) using the cor function in the R program (https://www.r-project.org). Genes with averaged RPM among replicates ≥1 were considered expressed. Principal component analysis (PCA) was performed to compare the log$_2$-transformed RPM values of the expressed gene profiles among cell/tissue-type and stages using the prcomp function in R. Hierarchical clustering of ARF and IAA sequences were carried out using GENE-E (https://software.broadinstitute.org/GENE-E/) with one minus Pearson correlation as the distance metric. Pairwise PCC values between the expression patterns of *ARF* and *Aux/IAA* genes were calculated with relative RPMs (RPM$^{geneX\ in\ sample-typeX}$/averaged RPM$^{geneX\ across\ all\ sample-types}$) using the cor function in R. DEGs among latitudinal sections were identified using an exact test provided in the edgeR software package[77], where genes were considered differentially expressed if the ratio was ≥1.5 or ≤0.66 and an adjusted *p* value (false discovery rate (FDR)) was <0.05.

**Gene annotation and functional enrichment analysis**. Gene descriptions were assigned according to the ITAG 2.4 loci descriptions (ftp://ftp.solgenomics.net/genomes/Solanum_lycopersicum/annotation/ITAG2.4_release/ITAG2.4_loci_gene_descriptions.txt). Gene ontology (GO) terms for *S. lycopersicum* were obtained from the PANTHER classification system version 10[34]. The GO enrichment analysis was performed for GO Biological Process Complete using the Bioconductor R library clusterProfiler[78], applying a hypergeometric test with FDR correction (adjusted *p* value < 0.05).

**Visualization of fruit anatomical structures**. Images of fruit sections or cell/tissue types were generated by hand tracing high-resolution photographs or light microscopy viewed pericarp sections, respectively. To create three-dimensional (3D) images of fruit, including internal structures, micro-computed tomography (micro-CT) was used to scan individual fruit at 10, 15, and 20 DPA (14.2, 16.9, and 26.8 μm/voxel resolution, respectively; Zeiss Xradia 520 Versa), and at 30 DPA, MG, Br, and Pk (50 μm/voxel resolution, GE Healthcare eXplore CT-120). Digital radiographs were reconstructed into 3D images using OsiriX software v.5.7 (Pixmeo SARL).

**Data integration into the TEA database**. Expression data and anatomical images were deposited in the TEA database (http://tea.sgn.cornell.edu)[15]. Spearman correlation coefficients of gene expression profiles were calculated using the RPM average values for all genes among all samples.

**Weighted gene co-expression network analysis**. Co-expression network modules were identified using averaged RPM values and the WGCNA package (v1.51) in R[33]. Genes with a low coefficient of variation of averaged RPM (CV < 1) among all sample types (cell/tissue types, different latitudinal sections, or developmental stages) were discarded and the remaining 12,662 genes were used for the analysis. The co-expression modules were obtained using the automatic network construction function (blockwiseModules) with default settings, except that the soft threshold (power) was 16, TOMType was signed, minModuleSize was 30, and mergeCutHeight was 0.25. Relative RPM values were averaged to summarize expression pattern of individual modules. A module eigengene (ME) value, which summarizes the expression profile of a given module as the first principal

component, was calculated for each individual module. Intermodular similarities were calculated as a PCC between ME values. ME-based gene connectivity (kME), a quantitative factor indicating the correlation strength of an individual gene in each module, was calculated using the signedKME algorithm as a PCC between the RPM expression levels of each gene and the ME of modules. The central hub genes in this study were defined as those with kME > 0.9 within the assigned module. To determine if there was a significant difference between the frequency of gene sets clustered in a particular WGCNA module and with relative frequency in the genome, a hypergeometric test was performed using the phyper function with FDR correction in R.

**Transmission electron microscopy.** Pericarp segments ($2 \times 2$ mm) were fixed in 4% paraformaldehyde/1% glutaraldehyde in 0.075 M Sorensen's buffer (pH 7.4; EMS, PA, USA) for 45 min at 4 °C. The fixed segments were washed with ice-cold 0.075 M Sorensen's buffer and then post fixed on ice with 0.75% osmium tetroxide in 0.075 M Sorensen's buffer for 90 min or overnight. The segments were washed, dehydrated in a stepwise acetone series, and embedded in Spurr's low viscosity resin (EMS, PA., USA). Sections (60–100 nm) were cut on a Leica Ultramicrotome, collected on 0.4% Formvar-coated grids, and viewed with a Zeiss Libra 120 TEM at 120 kV.

**BiFC assay.** Full-length cDNAs corresponding to *SlARF3* (Solyc02g077560), *SlARF4* (Solyc11g069190), and *SlIAA15* (Solyc03g120390) were cloned into the binary vectors pYFC43 and pYFN43[79] and introduced into *Agrobacterium tumefaciens* (GV2260). The constructs were agroinfiltrated into *Nicotiana benthamiana* leaves[79] without using a silencing suppressor and expression of the fusion proteins was examined 2 days later with a Leica TCS-SP5 confocal microscope (Leica Microsystems, Exton, PA USA) using a ×20 water immersion objective. YFP was excited with the blue argon ion laser (514 nm). Images processed using Leica LAS-AF software (version 2.6.0).

**Construction and transcript profiling of transgenic plants.** To create the *SlGRAS38* (Solyc07g052960) RNAi construct, a 381 bp fragment of the *SlGRAS38* transcript was amplified using cDNA from tomato fruit pericarp (cv Heinz 1706) using the GRAS_attB1 and GRAS_attB2 primers and recombined into the pHELLSGATE2 vector (kindly provided by CSIRO, Plant Industry, Canberra, Australia) using a BP Clonase enzyme kit (Invitrogen), according to the manufacturer's instructions. The construct was introduced into tomato (cv Ailsa Craig; AC) by *Agrobacterium*-mediated transformation using *A. tumefaciens* (LBA 4404)[53]. Primer sequences are listed in Supplementary Table 1.

Total RNA was extracted from total pericarp tissue at the Br and Br + 7 stages of AC (WT) and two $T_2$ *SlGRAS38*-RNAi lines using an RNeasy Mini Kit (Qiagen). qRT-PCR was carried out to analyze the expression of *SlGRAS38* with three biological replicates in technical triplicate using a *Power* SYBR Green RNA-to-C$_t$ 1-step kit (Applied Biosystems) in a 5 μL total reaction volume, with 5 ng RNA template, using an ABI PRISM 7900HT system (Applied Biosystems). *RIBOSOMAL PROTEIN L2* (*RPL2*; Solyc10g006580) was used as an internal control. Initial screening of $T_0$ *SlGRAS38*-RNAi lines was conducted using single Br-stage fruit and with *18S* as the internal control, but was otherwise performed as described above. Primer sequences are listed in Supplementary Table 1.

For RNA-Seq analysis of these samples, strand-specific libraries derived from the total RNA were constructed and sequenced, and read mapping was performed as above. Raw counts for each tomato gene were normalized to reads per kilobase million mapped reads (RPKM). A summary of the mapping data is shown in Supplementary Tables 2 and 3. Genes with averaged RPKM among replicates ≥1 were considered expressed. Significantly downregulated genes in the *SlGRAS38*-RNAi fruit pericarp compared to WT were identified using the edgeR package (FDR-adjusted *p* value <0.05, exact test) with ratio ≤0.5.

**Carotenoid and ethylene measurements.** Carotenoid extracts[80] from approximately 200 mg frozen pericarp were dried at reduced pressure and reconstituted in ethylacetate containing 50 μg/ml of diindolylmethane (internal standard). Carotenoids were separated and quantified using an ultra performance convergence chromatography (UPC$^2$) system and a $2.1 \times 150$ mm HSS C18 SB column packed with 1.8 μm diameter particles (Waters, Milford, MA). The column effluent was monitored between 250 and 500 nM. See Supplementary Table 4 for details. β-carotene was used to construct a 5-point calibration curve (10–140 ng/μl) and the relative concentrations of each carotenoid are reported in β-carotene equivalents.

Ethylene production by the fruit was measured with seven biological replications using a gas chromatograph (Hewlett-Packard 5890 series II) with standard curve method and normalized by fruit mass[14].

**McrBC-PCR.** Genomic DNA extracted from fruit of cv M82 plants was quantified using a Nanodrop (Thermo Fisher Scientific) and fractionated on a 1% agarose gel to check integrity. A total of 0.5 μg DNA was digested with 7.5 U McrBC (NEB) according to the manufacturer's recommendations for 6 h at 37 °C, or the same reaction without GTP as a negative control. PCR was performed using 40 ng McrBC-treated DNA. Thermal cycling conditions were: 95 °C for 2 min followed by 30–33 cycles of 30 s at 95 °C, 30 s at 55 °C and 1 min at 72 °C, and finally 5 min at 72 °C. Primer sequences are listed in Supplementary Table 1.

**Data availability**. The raw transcriptome sequences have been deposited into the NCBI sequence read archive (SRA) under accessions SRP109982 for fruit cell/tissue types and SRP109878 for the fruit pericarp from *SlGRAS38*-RNAi lines and the control. Transcriptome profiles can be accessed at the TEA (http://tea.solgenomics.net/) that integrated as a web-based bioinformatics tool in the Sol Genomics Network (SGN, https://solgenomics.net/). All other data supporting the findings are available in the paper and the Supplementary Information files.

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

## Acknowledgements

We acknowledge the Genomics Facility and the Imaging Facility of Cornell University's Biotechnology Resource Center (Institute of Biotechnology), and the BTI Plant Cell Imaging Center. This project was supported by grants to J.K.C.R., J.J.G., C.C., and Z.F. from the Plant Genome Research Program of the US National Science Foundation (IOS-

1339287), and to J.J.G. and Z.F. from the same program (IOS-1539831) and the US Department of Agriculture–Agricultural Research Service, to J.K.C.R. from the Agriculture and Food Research Initiative of the US Department of Agriculture (2016–67013–24732) and to Y. Shinozaki from the Japan Society for the Promotion of Science (16J00582).

## Author contributions

P.N., L.B.B.M., and S.I.S. contributed to tissue collection and separation; Y. Shinozaki and Y.X. performed LM and cDNA library construction; Y. Shinozaki, N.F.-P., Q.M., Y.Z., S.I. S., E.R.-M., Z.F., and L.A.M. contributed to bioinformatics and database development; P. N. performed BiFC assays; D.S.D. performed TEM analysis; D.E., Y. Shi, T.W.T., and K. C. contributed to transgenic plants analysis; D.E. performed epigenetic analysis. Y. Shinozaki, P.N., N.F.-P., D.E., T.W.T., C.C., Z.F., J.J.G., and J.K.C.R. wrote the manuscript. All authors reviewed and approved the final manuscript.

## Additional information

**Competing interests:** The authors declare no competing financial interests.

