## [Peer Review File · Nature Communications]

Reviewers' comments:

Reviewer #1 (Remarks to the Author):

The authors present a very detailed analysis of gene expression in developing and ripening tomato fruit, both in time as well in space, including tissue specificity as derived from the analysis of laser dissection-microscopy samples. They have integrated these data in a website, already accessible to the community, with great visualization and analysis power, such as for co-expression analysis. Tomato is the model crop for studying processes in fleshy fruit development and ripening, and the availability of these data as well as of the website is likely to be of great value for the research community specializing in this research area, as well as for a larger group of researchers. The results also raise the awareness, although probably already present somewhere in the back of our minds, that with analysis of whole fruit or whole pericarp gene expression only, valuable details get lost by the admixture of tissues developing at different rates and in part not identically at all. I believe the claims as stated in the abstract have been adequately supported by the analysis presented in the manuscript and supplementary data.

Although the manuscript is largely descriptive, it does give two examples of the application of the results to the formulation and testing of research hypotheses in two examples, the prediction of functional protein/protein-interactions in the ARF/IAA example and the functional analysis of a member of co-expression module, GRAS38 by RNAi.

With regard to the latitudinal gradient of gene expression during ripening, I think it is necessary to note and pay attention to the fact that the cultivar used here, M82, is in fact a uniform mutant (<http://tgrc.ucdavis.edu/%5C/Data/Acc/AccDetail.aspx?AccessionNum=LA3475>) and therefore not quite wild type in its ripening behavior. As the authors must be well aware, UNIFORM plays a role in establishing a gradient of chloroplast development leading to the “green shoulder” phenotype, which is known to affect basipetal ripening progression. Thus it may not be unlikely that the use of the mutation attenuates the gene expression gradient that is normally present. I understand the dilemma of choosing between a real wild type fruit and a representative of the most commonly used modern accessions (M82, MoneyMaker, MicroTom; Ailsa Craig being an exception), so I have no opinion there. However it would be good to briefly comment on this effect in the Discussion section with proper references, as readers may not always be aware of this.

Furthermore the manuscript is very well written, with an appropriate distribution between main

manuscript body and supplementary data. The latter are also quite complete in reporting the details of the expression analysis. I have no other remarks on the text.

Reviewer #2 (Remarks to the Author):

This study provides extensive gene expression profiling in ripening tomato that successfully demonstrated transcriptional dynamics in space and time within fruit. Some functional support for predictions is provided.

I however have reservations as to the analyses performed and their presentation:

Co-expression does not infer gene networks. This is correlations between the abundance of mature transcripts. Examining the topological landscape of correlations is a house of cards, stacking correlations on top of further correlations. Caution is needed as outlined below.

Line 213 – co-expression does not predict protein interaction. This is in fact the exception and not the rule, especially in light of <10% of transcripts being correlated with their corresponding proteins in yeast. This is not a surprising result and framing it as a prediction of co-regulation of mature transcripts is misleading and inaccurate.

Line 225 “A significant barrier to elucidating gene regulatory networks “ protein-protein interactions are not gene regulatory networks. These are protein-DNA interactions. The inaccuracy in the language here requires amendment.

Line 256 – there is no such thing as a hub gene in a correlation network. Hub implied causal control while there is none where correlations are involved.

Line 298 – again notion of hub needs to be removed.

WGCNA used to identify novel GRAS protein. Is this transcript simply correlated with RIN?

Line 353. SIDML2 is mentioned, then methylation at the RIN promoter shown, but no causal link is provided. Why was this not examined if mentioned?

Line 186 – why is the expression of these genes interesting?

What is concluded by having done WGCNA?

What are the main conclusions of this work?

Reviewer #3 (Remarks to the Author):

This manuscript reports on a set of RNA expression data for the developing tomato fruit using hand-dissected and laser-capture microdissected (LM) tissues to develop a spatiotemporal map. The manuscript further addresses a number of questions regarding the types of gene networks involved in the metabolic processes related to fruit quality and ripening, gene regulatory processes associated with fruit ripening, auxin response, and epigenetic regulation.

A key aspect of this manuscript is the extent of the individual biological samples collected, which has enabled the authors to generate a large number of co-expression networks (using WGCNA), used for the subsequent analysis of the above-mentioned processes—and presumably of other processes of interest to the community. This work is massive in its scope and contains at least 6 different questions/stories that flow from the spatiotemporal analysis. The authors clearly show the utility of the data and identify some key questions that can be followed up in subsequent analyses. The manuscript, largely based on existing technology (LM) and standard co-expression tools (WGCNA), is well documented in technical and analytical details. I have some “minor” concerns about the manuscript that I have indicated below.

The scale of the work has clearly challenged the authors to put together a cogent manuscript that would show the full extent of the work here. For example, the Introduction is nearly devoid of any specific and pertinent biological information to enable the reader to understand the context and extent of the analyses reported here—that is, there is no context described for the actual data presented in the paper. How would the different questions/threads be related to fruit development and ripening? The current introduction reads more like a justification for a grant proposal. I don't think anyone would have to be convinced of the utility and power of laser-capture methods (or simple hand dissection of tissues) for developing a spatiotemporal map of gene expression in any organism, and that the previous expression data with the whole-organ extraction methods should be reassessed. Similarly, the Discussion section is relatively thin on perspective and conceptualization. Associated with this, there seems to be no mention of any previous work utilizing a similar approach in other plant systems or developmental contexts.

With respect to the analysis of the genes expressed “specifically” in a particular cell or tissue (throughout), the authors seem to rely simply on the presence or absence of genes based on the original mapping and subsequent exclusion from a list of overlapping genes. I think the notion that a simple overlap test would identify specifically expressed genes without any quantitative measure of expression is non-rigorous and may lead to identification of genes with noisy expression patterns here. There does not seem to be a clear validation approach taken to support

the authors' conclusions in most cases mentioned in the manuscript.

Likewise, no data is shown to support the notion that non-detected total pericarp genes were in fact low-abundance pericarp cell-specific genes (lines 106-107, "LM allowed us to detect transcripts that were only present in low amounts in certain pericarp cell types, and that are overly diluted in total pericarp samples.") A graph with a quantitative analysis of these genes compared to the relevant sets would be useful here.

Reviewer #4 (Remarks to the Author):

This manuscript describes RNAseq analysis of multiple distinct tomato fruit tissues throughout development and ripening. The data are available in an online database that will be useful for tomato and fruit researchers studying any fruit process. This will be a great asset to determine the genes and gene interactions important for many fruit functions. The authors have also used many specific examples to demonstrate the usefulness of the data. Overall, this is a great example of using the newest technologies to create a database for the scientific community.

The manuscript is well-written and describes the objectives, results, methods and significance of the results in a concise and thorough manner. The following are suggestions for improving the manuscript:

RNA expression levels are not always correlated with protein or enzyme activity levels. A mention of this in the discussion is needed. The authors have shown examples with strong correlation between RNA and the products of enzyme activities; however, an example where this is not the case would demonstrate that sometimes we need to look beyond RNA levels.

Hyodo et al. (2013, PloS ONE) have published a study of pectin changes in many fruit tissues throughout development and ripening. Adding discussion of their results as compared to the RNAseq data would be another confirmation of the importance of the current data to tomato researchers

Were only two SIGRAS38 RNAi lines produced? Were progeny from transgenic lines lacking the transgene examined? These lines still have fairly high levels of GRAS expression, and carotenoid levels are only somewhat reduced. The differences in lycopene levels could easily be attributed to slight differences in ripening stage at the time of collection. Indeed, the ripening stage at 4d after breaker can vary greatly with environmental conditions. More independent transgenic lines grown under very controlled conditions would make this data more convincing.

Figure 1a. Is it possible to make the shaded areas clearer? I found it hard to see them.

Figure 2a, 2b, 5a. A different color gradient would be better. It is difficult to determine if white is no expression or no tissue. It also difficult to distinguish high expression (black) from the lines in the figure.

Line 382. It is deceptive to compare RNA blot analysis to RNAseq. Although rbcS3A was not

seen in total pericarp RNA blot analysis, it is detected in total pericarp in the RNAseq data presented in this paper. At 10DAP the rbcS3A was similar between total pericarp and most pericarp tissues with inner pericarp being on 10X higher. This section should be removed from the paper.

Were ethylene levels measured in the ripening fruit stages? Ethylene levels can be used to more accurately stage fruit tissue than days past breaker stage.

Abstract, lines 39-41. LM is mentioned, but hand dissection is not mentioned.

Shinozaki et al (NCOMMS-17-21129-T): response to reviewers' comments:

Reviewer #1 (Remarks to the Author):

1) With regard to the latitudinal gradient of gene expression during ripening, I think it is necessary to note and pay attention to the fact that the cultivar used here, M82, is in fact a uniform mutant

(<http://tgrc.ucdavis.edu/%5C/Data/Acc/AccDetail.aspx?AccessionNum=LA3475>) and therefore not quite wild type in its ripening behavior. As the authors must be well aware, UNIFORM plays a role in establishing a gradient of chloroplast development leading to the “green shoulder” phenotype, which is known to affect basipetal ripening progression. Thus it may not be unlikely that the use of the mutation attenuates the gene expression gradient that is normally present. I understand the dilemma of choosing between a real wild type fruit and a representative of the most commonly used modern accessions (M82, Moneymaker, MicroTom; Ailsa Craig being an exception), so I have no opinion there. However it would be good to briefly comment on this effect in the Discussion section with proper references, as readers may not always be aware of this.

- Response: We thank the reviewer for raising this issue. As shown in Nguyen et al. (2014, *Plant Cell* 26:585-601), GLK2 is not expressed along a "ripening" gradient and ripening of *u/u* fruit also starts from the stylar-side and then spreads to the stem-side. This suggests that the green shoulder phenotype and ripening gradients are largely independent. We confirmed that M82 also shows a latitudinal gradient indicated by pigmentation (Fig. 1 and Supplementary Fig. 1). However, we agree with the reviewer that it would be helpful to mention this phenomenon as it may cause confusion to some who are familiar with the *u* mutation, and accordingly have added relevant text to the results section in the middle of page 12.

Reviewer #2 (Remarks to the Author):

2) Co-expression does not infer gene networks. This is correlations between the abundance of mature transcripts. Examining the topological landscape of correlations is a house of cards, stacking correlations on top of further correlations. Caution is needed as outlined below.

- Response: We agree that the term "gene network" should be used carefully, since this can be interpreted as a broad definition referring to many types of molecular networks (as in Emmert-Streib et al., 2014 *Front Cell Dev Biol*), and have modified the text in a number of places to clarify the meaning. Nevertheless, it has been demonstrated in numerous studies that this is a powerful approach to identify transcript-transcript associations, which we have here defined as a type of gene network, and which we use to predict functionally associated genes.

3) Line 213 – co-expression does not predict protein interaction. This is in fact the exception and

not the rule, especially in light of <10% of transcripts being correlated with their corresponding proteins in yeast. This is not a surprising result and framing it as a prediction of co-regulation of mature transcripts is misleading and inaccurate.

- Response: We absolutely agree that transcriptional co-expression does not guarantee protein interactions. Nonetheless, co-expression data can provide a useful starting point to generate hypotheses and experimentally test functional protein interactions, as we illustrate in our study by identifying *SIARF4* and *SIIAA15* as putative interacting proteins based on tight co-expression, and then testing it using BiFC assays. This is particularly the case if combined with other data resources, as used in this study, such as proteome and cellular localization data. We note that many studies have shown that co-expression of transcripts corresponding to interacting proteins is often statistically significant (e.g. in Arabidopsis, Boruc et al., 2010 *Plant Cell*, cited in the first sentence of the related section on page 8).

4) Line 225 “A significant barrier to elucidating gene regulatory networks “protein-protein interactions are not gene regulatory networks. These are protein-DNA interactions. The inaccuracy in the language here requires amendment.

- Response: We have modified the text to clarify the meaning (see top of page 9).

5) Line 256 – there is no such thing as a hub gene in a correlation network. Hub implied causal control while there is none where correlations are involved.

- Response: We note that the term "hub gene" is ubiquitously used in WGCNA and other correlation network studies, including foundation publications in this area (e.g. *PLoS ONE* 8(4): e61505. doi:10.1371/journal.pone.0061505 and the primary tutorial from Horvath and Langfelder, “*Tutorial for the WGCNA package for R*”). The widely used definition of a hub gene, as defined by the authors of the original paper presenting the concept of WGCNA, is one with many connections or high connectivity to other genes within the module. Many studies have shown that highly connected hub genes can play important central roles in organizing the behavior of modules (e.g. Horvath et al., 2006, *PNAS*; Kang et al., 2011, *Nature*; and Parikshak et al., 2016, *Nature*). In short, we use the term “intramodular hub” as defined in Langfelder et al. (2008) and refer to the publication when discussing our work.

6) Line 298 – again notion of hub needs to be removed.

- Response: See response to 5) above.

7) WGCNA used to identify novel GRAS protein. Is this transcript simply correlated with RIN?

- Response: Yes, as we mention in the paper, both genes were clustered into the same co-expression module with high connectivity, meaning that their expression was highly correlated and that their expression patterns are representative of the assigned module. Based on this co-expression and the fact that GRAS38 function had not been previously defined, we sought to determine its function via repression in transgenic plants.

8) Line 353. *SIDML2* is mentioned, then methylation at the RIN promoter shown, but no causal link is provided. Why was this not examined if mentioned?

- Response: We appreciate the reviewer's pointing to our need to better justify why we examined the RIN promoter as a likely target for early demethylation events in the locule. Based on the pattern of *SIDML2* expression (i.e., beginning in the locule and moving outward to the pericarp), and given the demonstrated role of *SIDML2* in mediating ripening-associated promoter hypomethylation (Liu et al., 2015; Lang et al., 2017), we postulated that DNA demethylation would also occur first in the locule. Liu et al., (2015) used McrBC-PCR in *SIDML2* RNAi lines and showed that a region of the *RIN* promoter (the same region as p*RIN-2* in our experiment; manuscript Figure 6) is normally demethylated in WT at the fully ripe stage, but remains methylated in similarly staged *SIDML2* RNAi lines. Furthermore, Lang et al., (2017) demonstrated that additional regions of the *RIN* promoter are hypermethylated in *SIDML2* CRISPR knockout lines. These experiments provide evidence that the *RIN* promoter is a likely target of *SIDML2*-mediated demethylation. Thus, we examined the *RIN* promoter as a logical indicator and proxy for locule demethylation, which we indeed observed and reported here. We have modified the text in the last paragraph of the Results (page 13) to clarify why we selected the *RIN* promoter for analysis.

9) Line 186 – why is the expression of these genes interesting?

- Response: Identified genes that were co-expressed with *SICH5-2* in this study included the gene encoding the SIMYB12 transcription factor and 'unknown function' genes that are known to be induced by the Arabidopsis ortholog of SIMYB12. These data further support the deduced regulatory and functional association among the identified genes. Text has been added to this effect at the end of page 7.

10) What is concluded by having done WGCNA?

- Response: A total of 43 modules were identified, based on the co-expression pattern and association with biological functions, including known and new aspects of tomato fruit ontogeny. The text describing the WGCNA analysis and outcome is included in the Supplementary Note.

The spatiotemporal significance of processes associated with carbon metabolism and fruit ripening were further analyzed in subsequent stages of the study. We have modified the text in the corresponding parts of the Results section (the top of page 8) to clarify this.

11) What are the main conclusions of this work?

- Response: In this study we generated a comprehensive fruit transcriptome atlas at an unprecedented level of spatiotemporal resolution. We illustrate the value of these data in shedding new light on well-studied aspects in fruit biology, and in facilitating the development of hypotheses associated with multiple developmental programs. Several such hypotheses, for example, 1) regarding predictive power in defining interacting proteins (ARF and AUX/IAA), and 2) identifying candidate genes to be tested for function associated with ripening (GRAS38), and 3) predicting the activity of early ripening phenomena (DML2 demethylation), in addition to others, are presented. We believe these points are made clearly in the manuscript in its present form and are emphasized and summarized in the last two paragraphs of the discussion.

Reviewer #3 (Remarks to the Author):

12) The scale of the work has clearly challenged the authors to put together a cogent manuscript that would show the full extent of the work here. For example, the Introduction is nearly devoid of any specific and pertinent biological information to enable the reader to understand the context and extent of the analyses reported here—that is, there is no context described for the actual data presented in the paper. How would the different questions/threads be related to fruit development and ripening? The current introduction reads more like a justification for a grant proposal. I don't think anyone would have to be convinced of the utility and power of laser-capture methods (or simple hand dissection of tissues) for developing a spatiotemporal map of gene expression in any organism, and that the previous expression data with the whole-organ extraction methods should be reassessed. Similarly, the Discussion section is relatively thin on perspective and conceptualization. Associated with this, there seems to be no mention of any previous work utilizing a similar approach in other plant systems or developmental contexts.

- Response: The reviewer points out the massive scope of the analysis and summarizes our strategy to present the project, which was to highlight the value of high spatiotemporal resolution data (as Reviewer 1 notes, we have found that this is not always fully appreciated) using targeted set of questions related to key aspects of fruit development. A major 'take-home' is that the data sets shed new light on all these areas, even with regard to fundamental aspects of biology. The response to date from the community regarding our study has echoed the excitement that we hope this manuscript conveys.

13) With respect to the analysis of the genes expressed “specifically” in a particular cell or tissue (throughout), the authors seem to rely simply on the presence or absence of genes based on the original mapping and subsequent exclusion from a list of overlapping genes. I think the notion that a simple overlap test would identify specifically expressed genes without any quantitative measure of expression is non-rigorous and may lead to identification of genes with noisy expression patterns here. There does not seem to be a clear validation approach taken to support the authors’ conclusions in most cases mentioned in the manuscript.

- Response: Validation of ‘specificity’ is particularly difficult in extremely large data sets as ours, which consisted of 483 samples, resulting in study that is more comprehensive than most reports that target one, or a limited number of stages. There are various ways to define ‘specificity’ and our strategy was to address this in a couple of ways: one with a Venn diagram, which provides an initial overview; and another one with co-expression analysis by WGCNA, which provides major transcriptional patterns including spatial and incorporates that idea of temporal specificity. As the reviewer points out, gene sets shown in the form of a Venn diagram may be thought of as somewhat simplistic, and contain genes with more complex expression patterns, including those showing a distinct range of expressional levels depending on cell/tissue-types and developmental stages. Nevertheless, in the paper we noted significant differences in various comparisons, such as seeds and vascular tissues, which clearly indicated distinct transcriptional profiles in these cell/tissue-types, particularly taken together with the results of the PCA analysis. However, we have been careful to define what we mean by ‘specificity’, and in some case have used terms such as “predominant” or “dependent” for highly expressed genes in particular cell/tissue-types rather than “specific”, since most of those genes were actually also identified in most cell/tissue-types. We also note that such terminology has been suggested in several publications (e.g. Uhlen et al., 2015 *Science*). To help clarify this issue we have modified the text in first paragraph of page 7 and second paragraph of page 10.

14) Likewise, no data is shown to support the notion that non-detected total pericarp genes were in fact low-abundance pericarp cell-specific genes (lines 106-107, “LM allowed us to detect transcripts that were only present in low amounts in certain pericarp cell types, and that are overly diluted in total pericarp samples.”) A graph with a quantitative analysis of these genes compared to the relevant sets would be useful here.

- Response: We thank the reviewer for this suggestion. We have added to the revised manuscript a comparison of expression levels between genes detected only in pericarp cells/tissues via LM and those detected in total pericarp as suggested. This analysis confirmed that the genes detected only via LM showed lower expression (Supplementary Fig. 6a). Please also see the end of page 4. In addition, we added a figure showing the distribution of the expression levels among all samples (Supplementary Fig. 3 and middle of page 4).

Reviewer #4 (Remarks to the Author):

15) RNA expression levels are not always correlated with protein or enzyme activity levels. A mention of this in the discussion is needed. The authors have shown examples with strong correlation between RNA and the products of enzyme activities; however, an example where this is not the case would demonstrate that sometimes we need to look beyond RNA levels.

- Response: We have modified the Discussion section to emphasize this point (the top of page 14) including a discussion about transcript-protein level correlations.

16) Hyodo et al. (2013, PloS ONE) have published a study of pectin changes in many fruit tissues throughout development and ripening. Adding discussion of their results as compared to the RNAseq data would be another confirmation of the importance of the current data to tomato researchers

- Response: We thank the reviewer for this suggestion. We have modified the text on page 6 to include mention of the study by Hyodo et al. However, we have not commented in detail on the data presented in Hyodo et al., as we feel that there are some technical issues with their study. Hyodo et al. used only semi-quantitative RT-PCR to assess gene expression and it appears that in some cases, such as for PG2, the PCR amplification was not only totally saturated but also produced unexpected bands in mesocarp samples. We therefore believe that there are likely both qualitative and quantitative inaccuracies. Moreover, the immunolabeling assays that were presented by Hyodo et al. are not clear. The authors referred to developmental or latitudinal differences in labeling with the monoclonal antibodies LM19 and LM20, but the descriptions of the results in the text do not match. Nevertheless, they report an interesting result that the degree of methyl-esterified pectin in locular tissue was lower (45–50%) at the immature fruit stage than in other tissues, including pericarp and septum. We now mention this result in our paper, together with the results of an older study (Jones et al., 1997 *Plant Physiol*) showing that locular gel at the mature green stage is enriched with demethylesterified homogalacturonan, detected by the JIM5 monoclonal antibody.

17) Were only two SIGRAS38 RNAi lines produced? Were progeny from transgenic lines lacking the transgene examined? These lines still have fairly high levels of GRAS expression, and carotenoid levels are only somewhat reduced. The differences in lycopene levels could easily be attributed to slight differences in ripening stage at the time of collection. Indeed, the ripening stage at 4d after breaker can vary greatly with environmental conditions. More independent transgenic lines grown under very controlled conditions would make this data more convincing.

- Response: In addition to *SIGRAS38* RNAi lines 13 and 20, four additional transgenic lines were generated that had expression levels reduced to between 12 – 27% of WT (cv Ailsa Craig; AC)

in the T0 generation. Due to limited fruit set in the T0 generation, single Br-stage fruit were used to screen for *SIGRAS38* expression. *SIGRAS38* RNAi line 20 was profiled in the T1 generation and expression levels were found to be similar to that of line 13. The analysis of the six generated lines is now presented as Supplementary Figure 16. Because lines 13 and 20 produced the most fruit and seed, and had among the lowest expression in our preliminary screen, they were chosen for more in depth characterization. We have modified the text in the middle of page 11 and in the methods section to clarify that multiple lines were generated and shown to have reduced *SIGRAS38* expression, and that two were characterized in greater detail.

Azygous lines were identified in the T1 generation but we focused our characterization on the WT untransformed control due to limited greenhouse space and to allow growth of the entire population of transgenic and control plants at the same time. This was necessary to minimize developmental or environmental variation as noted below. While the experiments described here use AC as our WT, visual investigations of several T1 azygous individuals revealed no readily observable differences in ripening compared to AC

With regard to the carotenoid phenotype, we note that while the images of *gras-13*, *gras-20* and WT are at the Br+4 stage (manuscript Figure 5d), the carotenoid data is from pericarp at the Br+7 stage, as noted in the legend. As stated above, all plants were grown at the same time to minimize environmental and developmental variability. It is also noteworthy that at this later stage of development, when the fruit are fully ripe and red, any slight differences in the early progression of ripening are less likely to have influenced carotenoid content (manuscript Figure 5c). In addition, carotenoid content was also measured from these same lines at the over-ripe Br+15 stage, and the results showed the same overall trend. These additional data are now included in Supplementary Figure 16 to minimize any concern regarding the consistency of the carotenoid accumulation data. Observation of the same trend at a second and later stage of development strongly suggests that differences in carotenoid profile are due to *SIGRAS38* repression and not slight variation in ripening progression. We have further edited the text at the middle of page 11 to clarify that all lines were grown together and staged by tagging, and that similar carotenoid profile shifts at two separate fully-ripe stages further support a change in carotenoid accumulation resulting from reduced *SIGRAS38* gene expression.

18) Figure 1a. Is it possible to make the shaded areas clearer? I found it hard to see them.

- Response: The image has now higher resolution, and the shaded areas are clearer.

19) Figure 2a, 2b, 5a. A different color gradient would be better. It is difficult to determine if white is no expression or no tissue. It also difficult to distinguish high expression (black) from the lines in the figure.

- Response: The figures that we uploaded for review were relatively lower resolution than the original images that display the color scale clearer, and high resolution images will be provided. However, in addition, an option has now been added to the TEA website, called "Expression color scale" (http://tea.sgn.cornell.edu/expression_viewer/input), which deals with this exact

issue and allows the user to set the color scale and an appropriate range according to the gene(s) of interest. The screen capture images used in the paper were based on ranges that we feel best present the analysis of the genes in question and also reflect feedback from a survey group of users.

20) Line 382. It is deceptive to compare RNA blot analysis to RNAseq. Although rbcS3A was not seen in total pericarp RNA blot analysis, it is detected in total pericarp in the RNAseq data presented in this paper. At 10DAP the rbcS3A was similar between total pericarp and most pericarp tissues with inner pericarp being on 10X higher. This section should be removed from the paper.

- Response: The section has been removed.

21) Were ethylene levels measured in the ripening fruit stages? Ethylene levels can be used to more accurately stage fruit tissue than days past breaker stage.

- Response: We appreciate the reviewers concern. Our group has considerable experience in characterizing many aspects of ripening in M82 fruit, as well as other cultivars, including ethylene production. M82 fruit show a highly reproducible pattern of ethylene production during ripening, which is well correlated with fruit color change. Ethylene levels increase at, and after, the breaker stage, remain high during pigment formation and taper after the fully ripe stage, as reported in our prior (e.g. Nguyen et al., 2014 *Plant Cell*) and additional studies. Days to breaker can vary with the calendar; however, our considerable experience (including in some cases ethylene measurements) is that fruit grown as a crop together are very uniform in their developmental timing. In our study, all the harvested fruit were rigorously checked in terms of not only days past breaker, but also external color using a colorimeter and a color chart including detailed description, as shown in the methods section, as well as internal color of the locular tissue. We believe that these criteria provide an equally, if not more, accurate measure of ripening for tissue sampling, than ethylene production of the fruit or the tissues (which cannot be determined without incubation for one to several hours after harvest and ideally following a period of at least 8- 12 hours to ensure that wound-induced ethylene has dissipated). Indeed, one of the key questions that our study addresses is the nature of ripening and so we have very carefully defined the nature of the tissue in the methods section, such that it can be replicated by other groups.

22) Abstract, lines 39-41. LM is mentioned, but hand dissection is not mentioned.

- Response: Hand dissection has been added.

Reviewers' Comments:

Reviewer #1 (Remarks to the Author):

I have nothing further to comment. I am satisfied with the response to my earlier comments and the correction made to the new manuscript accordingly.

Reviewer #2 (Remarks to the Author):

The authors have addressed fundamental concerns of this nature and interpretation of this study using a series of written arguments.

Reviewer #3 (Remarks to the Author):

The revised version of the manuscript does address two of my key issues from the previous version. The exception is addressing my criticism regarding the relatively superficial writing of the intro and discussion, and a lack of sufficient citation of similar studies (point 12 in the author rebuttal letter). The authors' response is rather a non-response, citing Reviewer #1 and the community's response in support of why these sections should not be revised! On this issue, I defer to the editor to decide if the authors' response is satisfactory.

Reviewer #4 (Remarks to the Author):

The authors have done an excellent job of addressing the concerns of the reviewers. I do not have any further questions for the authors.

Shinozaki et al (NCOMMS-17-21129-A): response to reviewers' comments:

Reviewer #3 (Remarks to the Author):

The revised version of the manuscript does address two of my key issues from the previous version. The exception is addressing my criticism regarding the relatively superficial writing of the intro and discussion, and a lack of sufficient citation of similar studies (point 12 in the author rebuttal letter). The authors' response is rather a non-response, citing Reviewer #1 and the community's response in support of why these sections should not be revised! On this issue, I defer to the editor to decide if the authors' response is satisfactory.

- Response: We thank the reviewer for the additional input and have modified the text in both the introduction and discussion sections accordingly. In the introduction, we now emphasize more the importance of the high-resolution expression data in the context of fruit biology and the biological questions that we address in this study. As requested, we have also added citations of similar transcriptome analyses of other plant organs. Similarly, in the discussion, we added text related to the perspective of future studies using our datasets.